# Natural Infection and Vertical Transmission of Zika Virus in Sylvatic Mosquitoes *Aedes albopictus* and *Haemagogus leucocelaenus* from Rio de Janeiro, Brazil

**DOI:** 10.3390/tropicalmed6020099

**Published:** 2021-06-11

**Authors:** Jeronimo Alencar, Cecilia Ferreira de Mello, Carlos Brisola Marcondes, Anthony Érico Guimarães, Helena Keiko Toma, Amanda Queiroz Bastos, Shayenne Olsson Freitas Silva, Sergio Lisboa Machado

**Affiliations:** 1Laboratório Diptera, Instituto Oswaldo Cruz (FIOCRUZ), Manguinhos 21040-360, Brazil; cecilia.mello@ioc.fiocruz.br (C.F.d.M.); anthony@ioc.fiocruz.br (A.É.G.); amanike2@hotmail.com (A.Q.B.); shayenneolsson@gmail.com (S.O.F.S.); 2Programa de Pós-Graduação em Biologia Animal, Instituto de Biologia, Universidade Federal Rural do Rio de Janeiro, Seropédica 23890-000, Brazil; 3Departamento de Imunologia e Parasitologia de Microbiologia, Centro de Ciências Biológicas, Universidade Federal de Santa Catarina, Florianópolis 88040-900, Brazil; cbrisolamarcondes@gmail.com; 4Laboratório de Diagnóstico Molecular e Hematologia, Universidade Federal do Rio de Janeiro, Rio de Janeiro 21941-901, Brazil; hktoma@globo.com; 5Programa de Pós-Graduação em Medicina Tropical, Instituto Oswaldo Cruz (FIOCRUZ), Manguinhos 21040-360, Brazil

**Keywords:** Rio de Janeiro, *Haemagogus leucocelaenus*, *Aedes albopictus*, Zika virus, yellow fever virus

## Abstract

Zika virus (ZIKV) was recently introduced into the Western Hemisphere, where it is suspected to be transmitted mainly by *Aedes aegypti* in urban environments. ZIKV represents a public health problem as it has been implicated in congenital microcephaly in South America since 2015. Reports of ZIKV transmission in forested areas of Africa raises the possibility of its dispersal to non-human-modified environments in South America, where it is now endemic. The current study aimed to detect arboviruses in mosquitoes collected from areas with low human interference in Rio de Janeiro, Brazil. Using a sensitive pan-flavivirus RT-PCR, designed to detect the NS5 region, pools of *Ae. albopictus* and *Haemagogus leucocelaenus*, were positive for both ZIKV and yellow fever (YFV). Virus RNA was detected in pools of adult males and females reared from field-collected eggs. Findings presented here suggest natural vertical transmission and infection of ZIKV in *Hg. leucocelaenus* and *Ae.*
*albopitcus* in Brazil.

## 1. Introduction

The family *Flaviviridae* contains four genera, including *Flavivirus*, a genus that contains over 50 viruses, with 13 being already reported in Brazil. Among the most important flaviviruses circulating in Brazil are yellow fever (YF), dengue (DENV), Zika (ZIKV) and West Nile viruses. 

Yellow fever has been a major health problem since the first epidemic occurred in Recife in 1685 [1]. This disease was essentially eliminated from urban areas by the early 20th century; however, in the 1920s, sylvatic transmission was detected in Colombian forest areas. Sylvatic transmission among non-human primates by forest mosquitoes was first identified and verified in Brazil in the Chanaan Valley, located in the state of Espirito Santo, southeast Brazil [2].

ZIKV, first identified in a forest in Uganda, has recently spread to Asia and the Pacific islands, and later to the Americas, where it caused explosive outbreaks in Brazil. ZIKV detection in the Western Hemisphere has caused serious concern owing to its association with fetal microcephaly [3,4,5]. Although mostly transmitted in urban environments by *Aedes aegypti* and possibly *Aedes albopictus*, it has been detected in many other species of mosquito [6]. The first autochthonous case of ZIKV infection in Brazil was diagnosed in May 2015 [7]. Its circulation has since been confirmed in all 26 states and federal districts of the country [8]. The importance of non-human primates for the maintenance of ZIKV in South America remains unknown. However, a few reports in Brazil suggest that non-human primates have been exposed to ZIKV [9,10,11,12]. 

There is a current outbreak of sylvatic yellow fever in Brazil that probably started at the end of 2016. The first cases were reported in the state of Minas Gerais, but YFV has since spread to the states of Espírito Santo, São Paulo and Rio de Janeiro, with recent cases appearing in the southern states of Paraná and Santa Catarina. According to a WHO report, as of April 2017, YFV (observed as either epizootic or human cases of yellow fever) continues to expand its distribution toward the Atlantic coast of Brazil to areas not previously considered to be at risk of transmission [13].

The main genera of mosquitoes transmitting sylvatic YFV are *Haemagogus* and *Sabethes*. In southeast Brazil during the present epidemic, *Hg. leucocelaenus* and *Hg. janthinomys* have been implicated as the important vectors [14]. *Haemagogus* are sylvatic mosquitoes with diurnal activity; they are mostly acrodendrophilic and mainly found in densely forested areas [15]. *Haemagogus leucocelaenus* is the species most frequently found in Brazil, and it is considered a primary vector for sylvatic YFV in southeastern Brazil. A study conducted in the northeast region of Brazil detected DENV-1 by reverse transcriptase polymerase chain reaction (RT-PCR) in *Hg. leucocelaenus* from Coribe, state of Bahia, suggesting the exposure of a sylvatic mosquito species to an arbovirus maintained in endemic urban transmission cycles in Brazil. These findings highlight the importance of arbovirus surveillance at the human–animal interface in Brazil [16,17].

In Africa, some studies have shown evidence of ZIKV exposure different orders of mammals, including non-human primates, birds and reptiles [18,19]. In Brazil, exposure of non-human vertebrates to ZIKV has been reported in different regions of the country. Capuchin monkeys and marmosets captured between June 2015 and February 2016, tested positive by real-time RT-PCR in the northeast region of the country [12]. More recently, neutralizing antibodies for ZIKV were detected in various domestic species, including goose, cattle, chicken, horse, dog and sheep, and also in a captive white-cheeked spider monkey from Mato Grosso and Mato Grosso do Sul states, Central-West Region, Brazil [9]. ZIKV RNA was also detected in carcasses of non-human primates during an epizootic outbreak of yellow fever in the southeast region. Positive animals were from the states of São Paulo and Minas Gerais [20].

Despite some evidence of exposure of non-human vertebrates to ZIKV in Brazil, evidence of ZIKV in sylvatic vector populations is scarce. Recently, an investigation into ZIKV in roughly 23,000 mosquitoes of 62 species from Central-West Region of Brazil found no positive results [9]. 

Ovitraps are important tools that make it possible to determine the presence of species of mosquito vectors of etiological agents through the eggs deposited in it [21,22]. 

Thus, the main objective of the present study was to detect ZIKV in vector populations in preserved forested areas located near active human transmission areas of the state of Rio de Janeiro.

## 2. Results

A total of 8086 mosquito eggs were obtained between September 2018 and March 2019. Of these, 3662 (55.1%) adults emerged, and 924 distributed in 70 pools were tested for flaviviruses.

Pools were separated by species, sex and ovitrap. Each pool had three to 33 mosquitoes. Six pools from two species were positive for flavivirus by RT-PCR and submitted to nucleotide sequencing. The minimum infection rate was not established due to the number of insects collected being lower than 1000.

When compared to known sequences in NCBI Blast, three pools were identified as YFV and three as ZIKV. When comparing the obtained sequences with those deposited in GenBank, they exhibited 94–98% similarity to ZIKV and YFV (Table 1). Nucleotide sequences from the NS5 segment of flaviviruses obtained in the present study were deposited in GenBank with the accession numbers MK972825, MK972826, MK972827, MK972828 and MK972829. We also sequenced our positive controls before sequencing our samples (data not shown). The sequences of the positive controls and positive samples were aligned using Geneious V10.2.4 software, resulting in the mean of 23.6% of identical sites and 82.3% of pairwise homology.

## 3. Discussion

The NS5 region of flaviviruses belongs to the last part of the open reading frame (ORF), encoding the largest and highly conserved protein. NS5 region was targeted in this study because it is a conserved region that has been used previously for flavivirus detection [7,23,24,25,26]. Through sequencing, we identified ZIKV and YFV RNA in two sylvatic mosquito species (Table 1). Sylvatic YFV is usually found in *Ae. albopictus*, *Haemagogus leucocelaenus* and *Hg. janthinomys* [27,28,29,30,31,32] and circulates in forests within non-human primates as amplifying hosts. This is one of the first detections of ZIKV RNA in wild mosquitoes present in forest environments of the Americas. *Haemagogus leucocelaenus* and *Ae. albopictus* are adapted to both sylvatic and urban environments and are considered epidemiologically important mosquitoes involved in the arbovirus transmission cycle. *Aedes albopictus* has already been reported as a natural ZIKV vector in several countries [26,33,34,35]. The presence of ZIKV RNA in *Hg. leucocelaenus*, an acrodendrofilic species, suggests a sylvatic maintenance cycle of ZIKV in Brazil. Laboratory studies have demonstrated the ability of *Ae. albopictus* to transmit ZIKV. Furthermore, the finding of YFV RNA in *Ae. albopictus* mosquitoes in this area suggests the risk of spillover from forest to human-modified environments [36,37].

*Haemagogus leucocelaenus* can feed on a wide range of vertebrates. Birds can be a common blood source for this species. However, these mosquitoes can also feed on several mammalian species present in forest environments, as shown in studies conducted in the Rio de Janeiro and Goiás states, Brazil [38]. *Aedes albopictus* feeds mostly on mammals, and, although it prefers human blood, it is widely distributed in the absence of humans [39,40].

*Aedes albopictus* was also found to be a vector for two other arboviruses, CHIKV and DENV, elsewhere [41]. In addition, through laboratory tests of co-infection and super-infection, the possibility of simultaneous transmission of these two viruses to humans has been demonstrated [42]. Our results indicated a possible sylvatic cycle for ZIKV in South America. *Aedes albopictus* inhabits both forested and peridomestic environments in Rio de Janeiro [43]; therefore, there is a high likelihood of viral transport between these habitats.

Mosquitoes in the states of Ceará and Bahia, where ZIKV has been found in mammals, should also be studied [12,44]. Although a small number of mosquitoes tested negative in the current study, *Hg. janthinomysis* is an important YFV vector in the canopy and could possibly transmit ZIKV [45]. This species was found in an urban forest (Parque Dois Irmãos) in Recife, Pernambuco, Brazil, a city highly endemic to ZIKV; for this reason, other arboviruses (DENV and CHIKV) should similarly be studied for natural infection [46].

The finding of ZIKV in mosquitoes reared from eggs obtained under natural conditions indicates vertical transmission of this virus. The role of vertical transmission in the maintenance of both YFV and ZIKV remains unclear, but suggests enzootic maintenance of these viruses in sylvatic cycles. Vertical transmission in vector mosquitoes has already been reported both naturally and experimentally. In 2016, the vertical transmission of the Zika virus in *Ae. aegypti* larvae was detected for the first time, found in the field under natural conditions [47]. In general, YFV enzootic transmission occurs mainly in forests, infecting humans entering the forest for activities such as logging, fishing and hunting. In the case of *Hg. leucocelaenus*, which can range beyond forest environments, humans of both sexes and various ages living near forests have been infected; thus, transmission does not appear to be limited to people engaged in forest-associated work [44]. 

YFV-infected *Hg. leucocelaenus* were found at ground level during an outbreak in Rio Grande do Sul state (Brazil) between 2008 and 2009 [48]. This supports our view that these mosquitoes can be a risk for the transmission of an arbovirus to human beings. An evaluation of the vector competence of *Hg. leucoceleanus* challenged orally and inoculated intrathoracically with ZIKV allowed the detection of the spread of this arbovirus in *Hg. leucocelaenus*. However, this detection was observed at very low rates [45].

The occurrence of YFV in natural conditions demonstrates its possibly circulating in Atlantic forest areas of the municipality of Casimiro de Abreu, Rio de Janeiro state.

Evidence of active sylvatic YFV transmission in the nature reserves studied here and the abundance of the main mosquito vector for this virus in Brazil indicates the need for active YFV surveillance in communities adjacent to forests. Forests near human-modified areas where arboviruses have been found, such as urban forests (e.g., Tijuca in Rio de Janeiro, Buraquinho in João Pessoa and Dois Irmãos in Recife) should be prioritized.

Mosquitoes adapted to urban environments, most commonly *Ae. aegypti*, transmit YFV and ZIKV between humans. Because both viruses can be transmitted by several mosquito species, spillback to preserved forests with wild amplifying vertebrates and mosquitoes should considered. However, if not studied, such sylvatic cycles will remain hidden, and low levels of ZIKV or YFV antibody reactivity of primates near urban areas [9,11] should not discourage additional studies in such areas.

## 4. Materials and Methods

### 4.1. Ethics Statement

All research was performed in accordance with scientific license number 44333 provided by the Ministry of Environment (MMA), Chico Mendes Institute of Biodiversity Conservation (ICMBio), Biodiversity Information and Authorization System (SISBIO). Mosquitoes were collected with the consent and cooperation of property owners, householders, and local authorities. All members of the collection team were vaccinated against YFV and aware of the potential risks in the areas under study.

### 4.2. Study Areas

Sampling sites were selected from forested areas near human transmission regions in the state of Rio de Janeiro, Southeast Brazil. Atlantic Forest fragments in the municipalities of Casimiro de Abreu and Nova Iguaçu were selected for their susceptibility to arbovirus transmission (Figure 1). The region was affected by a recent severe yellow fever outbreak in 2016–2018 [17].

The municipalities of Casimiro de Abreu and Nova Iguacu are, respectively, 140 and 30 km distant from the city of Rio de Janeiro. The main land cover in the region is Atlantic forest vegetation, with dense ombrophilous sub-mountain forests in moderate and advanced stages of regeneration. The region, located in the hydrographic basin of São João River, is situated in the intertropical zone (at low latitudes) and its climate is predominantly humid tropical [49]. The region is highly influenced by the Atlantic Ocean, and presents an average temperature of 26.8 °C, relative humidity of 56% and 1200 mm of precipitation annually. The highest levels of rainfall occur from October to March. 

### 4.3. Mosquito Sampling

Mosquito eggs were collected from ovitraps placed in three sampling sites from September 2018 to March 2019. The sites were located in the Três Montes Farm (A) and Três Morros Private Reserve of Natural Heritage (B), both in the Casimiro de Abreu, and Sítio Boa Esperança (C) in Nova Iguacu.

Fifteen ovitraps were set in each sampling site. In Casimiro de Abreu, the distance between traps was approximately 300 m, and that between traps in the Nova Iguacu was 1000 m. The distance between Nova Iguacu and Casimiro de Abreu was approximately 137 km.

Monitoring was performed using oviposition traps consisting of a 1L-capacity black container without a lid that resembled a plant vase, which contained four wooden oviposition paddles (Eucatex plates) measuring 2.5 cm × 14 cm, vertically held inside the trap by a clip. Natural water and remains of leaves and animals found on the forest soil were added into the trap to generate an ecosystem similar to the natural one [50,51,52]. 

The paddles in the traps were examined in the laboratory for eggs every two weeks for 14 weeks during the seven months of research. In the laboratory, egg-positive paddles were immersed in white trays filled with dechlorinated water at 29 ± 1 °C to allow the eggs to hatch. After three days, the paddles were removed from the water, left to air dry for another three days, and the hatched larvae were enumerated and reared as previously described [53]. Adults were identified to species level by morphological characters using dichotomous keys [15,54]. Specimens were grouped by species, trap and date of collection and stored at −80 °C prior to RNA extraction [55]. 

### 4.4. RNA Extraction

Males and females were separated and prepared in pools of 3–33 mosquitoes for maceration. Viral RNA was extracted from each pool under BSL2 laboratory conditions following the manufacturer’s instructions, using an MN Nucleo Spin RNA kit (Macherey-Nagel GmbH & Co. KG, ref. 740955.250, Quezon City, Phillipines) and cDNA was immediately synthetized using a Hi-capacity RNA-to-DNA kit (Applied Biosystem, ref. 4388950, Foster city, CA, USA), according to the manufacturer’s instructions. DNA was quantified using a Denovix DS-11+b Quantifier (DeNovix Inc., Wilmington, DE, USA) and maintained at −20 °C until it was tested for flaviviruses.

### 4.5. PCR for Flaviviruses 

For flavivirus detection, a conventional RT-PCR protocol using primers designed to amplify a ~260 nucleotides sequence of NS5 region was used, as previously described [56]. To control our amplification, we prepared one tube for each positive and negative control. As positive controls, we used cDNA obtained from Yellow Fever 17D, Zika virus and West Nile virus RNA, and as negative controls, we used cDNA from the Chikungunya virus (CHKV), mosquitoes that were not infected with any arbovirus and water.

The reaction mixture comprised 1× PCR buffer, 1.5 mM MgCl_2_, 10 pmol Pan-Flavi Forward primer (5′-TAC AAC ATG ATG GGG AAR AGA GAR AA-3′), 10 pmol Pan-Flavi Reverse primer (5′-GCW GAT GAC ACM GCN GGC TGG GAC AC-3′), 1.0 U of DNA polymerase (Thermo Fisher Scientific, 168 Third Avenue, Walthan, MA 02451, USA), 1.5 mM MgCl_2_, 1× Buffer (100 mM Tris-HCl pH 8.8, 500 mM KCl, 0.8%, Nonidet P40) and 0.2 mM dNTPs in a 25 µL reaction volume. The cycling conditions were 94 °C for 5 min, 35 cycles at 94 °C for 30 s, 55 °C for 30 s, 72 °C for 30 s and a final extension at 72 °C for 5 min.

PCR products were visualized after electrophoresis on 1.5% agarose gels in 1× TBE (Trizma, boric acid, EDTA) buffer under 77 light (260 nm) after ethidium bromide staining. The expected amplified fragments ranged 200–300 bp in size and were purified using a Cellco PCR purification kit (Cellco Biotec do Brasil Ltda, cat. # DPK-106L, São Carlos, SP, Brazil).

### 4.6. Nucleotide Sequencing

Sequencing was performed by Fiocruz nucleotide sequencing center, using an ABI 3730 DNA Analyzer (Applied Biosystems^®^, Foster City, CA, USA), as previously described [26]. Approximately 10–40 ng of purified PCR product was sequenced on an ABI 3730 DNA Sequencer following the BigDye Terminator v.3.1 Cycle Sequencing protocol. The sequences were then analyzed using Geneious R10 (Biomatters, v.10.2.6), and resulting contigs were compared with reference sequences using NCBI Blast (Basic Local Alignment Search Tool, at https://blast.ncbi.nlm.nih.gov/Blast.cgi accessed on 27 August 2020)

## 5. Conclusions

Results presented here corroborate the warnings rearing the transmission of ZIKV from urban to forest environments [19], making eradication of the virus from the continent highly improbable and reinforcing the need for the control of urban mosquitoes and the development of an effective vaccine. Owing to the evidence regarding the possible natural infection and vertical transmission of YFV and ZIKV found in *Hg. leucocelaenus* and *Ae. albopictus* in forested areas at Casimiro de Abreu and in Nova Iguaçú, Rio de Janeiro state, Brazil, active and constant entomological surveillance in the region is recommended to prevent the spread of these viruses to other sylvatic or urban areas.

## Figures and Tables

**Figure 1 tropicalmed-06-00099-f001:**
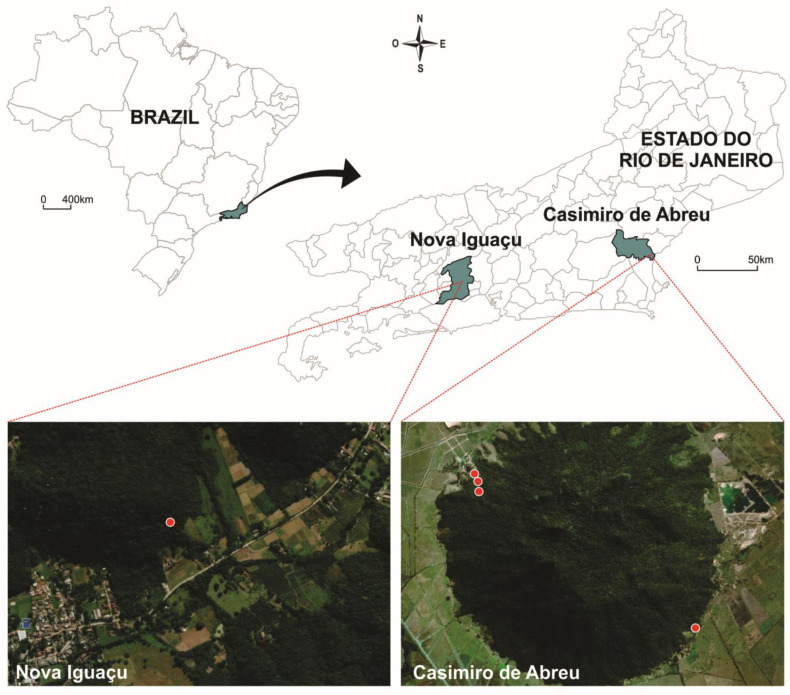
Map of the mosquito collection sites for this study. Red: Sampling sites where yellow fever (YFV) and Zika virus (ZIKV) positive mosquitoes were collected in primary forests in the Brazilian, municipalities of Nova Iguaçu and Casimiro de Abreu, state of Rio de Janeiro, Brazil. Maps were prepared in ArcGIS PRO (URL: https://pro.arcgis.com/en/pro-app/. Accessed: 6 May 2019) and edited in CorelDRAW Graphics Suite X7.

**Table 1 tropicalmed-06-00099-t001:** Detection of yellow fever virus and Zika virus in *Aedes albopictus* and *Haemagogus leucocelaenus* in primary forests in the Brazilian state of Rio de Janeiro, Brazil.

Pool ID *	Mosquito Species	Sex	Total Mosquitoes	Month/Year Collected	Geographic Coordinates	Trap Identification	Pan-Flavivirus PCR Result	Sequence Match
**43**	*Ae. albopictus*	♀	32	Jan/19	22°33′01.3″ S 42°00′52.7″ W	TMPRNH -32 *	Positive	Zika virus
**45**	*Ae. albopictus*	♂	2	Oct/2018	22°31′40.1″ S 42°02′58.6″ W	TMF -2 *	Positive	Zika virus
**54**	*Hg. leucocelaenus*	♀	6	Jan/19	22°35′11.98″ S 43°24′34.12″ W	Tinguá *	Positive	Zika virus
**62**	*Ae. Albopictus*	♀	9	Oct/2018	22°31′43.9″ S 42°02′56.8″ W	TMF -3 *	Positive	Yellow fever virus
**64**	*Hg. leucocelaenus*	♂	3	Sep/2018	22°31′49.5″ S 42°02′56.3″ W	TMF -7 *	Positive	Yellow fever virus
**65**	*Hg. leucocelaenus*	♂	5	Oct/2018	22°31′49.5″ S 42°02′56.3″ W	TMF -7 *	Positive	Yellow fever virus

* Pool ID, identification number for the processed mosquito pool. TMF, Três Montes Farm, sites 2, 3 and 7; TMPRNH, Três Morros Private Reserve of Natural Heritage, site 32; and Tinguá, Sitio Boa Esperança, Nova Iguaçu.

## Data Availability

Not applicable.

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
