# Peer review of "Natural Infection and Vertical Transmission of Zika Virus in Sylvatic Mosquitoes *Aedes albopictus* and *Haemagogus leucocelaenus* from Rio de Janeiro, Brazil"

_tropicalmed, 2021, doi:10.3390/tropicalmed6020099_

Round 1

Reviewer 1 Report

This paper investigates and reports on the current state of vertical transmission and maintenance of Zika virus in a population of non human blood feeding mosquitoes in the forest. I think that the paper presents robust and important finding and a study worth publishing. I would like to point out some minor things to make the manuscript easier to understand.

Introduction

L85 delete <trap>

Results

L92-94,  How do the authors deal mosquitoes larvae?

L 96. Five should be SIx?

L 100. Two should be Three?

Pool ID is given to two samples of two different date samples.

According to the MM, samples collected in different date should be given different sample ID.

L117-118. Please add more specific explanation.

L125-126.  Could the authors reconsider the text? I could not understood.

L152. Could the authors reconsider the text? which supports,, The front and back parts are not connected. 

L161.  ,,should be prioritized.

L221-222 With this method, only eggs that are resistant to drought can be collected. If the purpose is to collect only drought-tolerant eggs from the first place, please explain that this collection method does not collect all mosquito species but dose collect some intended species in the study?

L218-219. The authors should describe more how they collect eggs without forcing the readers to read a book and two papers. The size of the container not only the volume. How much of water, what kind of water was put or only rain water? We can not know any detail with the present description. 

References

Please describe species names in Italics. 

There are many places where words that should be lowercase in the paper titles are in uppercase. Please check every references carefully and make the style uniform.

Author Response

We are thankful for the comments and suggestions from anonymous reviewers that helped improve our manuscript. The new version of this manuscript has been read and edited by an English-speaking collaborator.

Reviewer: 1

We thank Reviewer #1 for the critical and valuable comments that contributed to the new revised version of the manuscript.

Review Report Form

Open Review

(x) I would not like to sign my review report

( ) I would like to sign my review report

English language and style

( ) Extensive editing of English language and style required

( ) Moderate English changes required

( ) English language and style are fine/minor spell check required

(x) I don't feel qualified to judge about the English language and style

Yes      Can be improved        Must be improved      Not applicable

Does the introduction provide sufficient background and include all relevant references?

(x)       ( )        ( )        ( )

Is the research design appropriate?

(x)       ( )        ( )        ( )

Are the methods adequately described?

( )        ( )        (x)       ( )

Are the results clearly presented?

( )        (x)       ( )        ( )

Are the conclusions supported by the results?

(x)       ( )        ( )        ( )

Comments and Suggestions for Authors

This paper investigates and reports on the current state of vertical transmission and maintenance of Zika virus in a population of non human blood feeding mosquitoes in the forest. I think that the paper presents robust and important finding and a study worth publishing. I would like to point out some minor things to make the manuscript easier to understand.

Introduction

L85 delete <trap>

Answer: Thank you for noticing this error. We have corrected the text as suggested.

Results

L92-94. How do the authors deal mosquitoes larvae?

Answer: The larvae were collected and taken into the lab, where they were kept in small polyethylene cups containing water from the same ovitrap from which they were collected. They were fed with TetraMint (Tetra, Blacksburg, VA) fish food and reared until reaching the adult stage. Adults were then identified through direct observation of morphological characteristics under a stereoscopic microscope, using the dichotomous keys proposed by Arnell (1973) and Forattini (2002). 

References:

Arnell, J. H. 1973. Mosquito studies (Diptera, Culicidae) XXXII. A revision of the genus Haemagogus. Contrib Am Entomol Soc. 10: 1–174.

Forattini O. P. 2002. Culicidologia Médica. vol 2: Identificação, biologia, epidemiologia.

L 96. Five should be SIx?

Answer: Thank you for your observation. The incorrect number is now corrected to SIX in the text.

L 100. Two should be Three?

Answer: Yes, thank you, this wrong number has also been corrected to THREE.

Pool ID is given to two samples of two different date samples.

Answer: Thank you for this observation. We have corrected this in the manuscript. Once the pools were analyzed separately, we added another line (65) in Table 1.

According to the MM, samples collected in different date should be given different sample ID.

Answer: That´s correct. We have now changed this information in the manuscript as suggested.  

L117-118. Please add more specific explanation.

Answer: This sentence was corrected in the manuscript for better clarification, as follows:

“This is one of the first detections of ZIKV RNA in wild mosquitoes present in forest environments of the Americas. Haemagogus leucocelaenus and Ae. albopictus are adapted to both sylvatic and urban environments and are considered epidemiologically important mosquitoes involved in the arbovirus transmission cycle. Aedes albopictus has already been reported as a natural ZIKV vector in several countries.”

L125-126.  Could the authors reconsider the text? I could not understood.

Answer: We have corrected the sentence in the manuscript as follows:

Haemagogus leucocelaenus can feed on a wide range of vertebrates. Birds can be a common blood source for this species. However, these mosquitoes can also feed on several mammal species present in forest environments, as shown in studies conducted in the Rio de Janeiro and Goiás states, Brazil.”

Alencar, J.; Marcondes, C.B; Serra-Freire, N.M; Lorosa, E.S.; Pacheco, J.B.; Guimarães, A.É. Feeding patterns of Haemagogus capricornii and Haemagogus leucocelaenus (Díptera: Culicidae) in two Brazilian states (Rio de Janeiro and Goiás). J. Med. Ento-mol. 2008, 45, 873–876, doi:10.1603/0022-2585(2008)45[873:FPOHCA]2.0.CO;2.

L152. Could the authors reconsider the text? which supports. The front and back parts are not connected.

Answer: We changed the structure of the text as suggested.

“YFV-infected Hg. leucocelaenus were found at ground level during an outbreak in the Rio Grande do Sul (Brazil) between 2008 and 2009. This supports our view that these mosquitoes can be a risk for the transmission of an arbovirus to human beings. An evaluation of the vector competence of Hg. leucoceleanus challenged orally and inoculated intrathoracically with ZIKV allowed the detection of the spread of this arbovirus in Hg. leucocelaenus. However, this detection was observed at very low rates.”

L161.  ,should be prioritized.

Answer: We changed the text as suggested.

L221-222 With this method, only eggs that are resistant to drought can be collected. If the purpose is to collect only drought-tolerant eggs from the first place, please explain that this collection method does not collect all mosquito species but dose collect some intended species in the study?

Answer: Ovitraps are an extensively used method for collecting and surveying various Culicidae species, not just those with drought-resistant eggs. All species that oviposit near the water surface can be collected with this trap (Moriya 1974, Tsuda et al. 1994, Navarro and Machado-Allison 1995, Yanoviak 1999, Yanoviak 2001, Ritchie et al. 2004). These eggs are then immersed in white trays filled with dechlorinated water at 29 ± 1°C to allow the eggs to hatch. Larvae that emerge from these hatched eggs are fed with TetraMint (Tetra, Blacksburg, VA) fish food and monitored daily until the immature mosquitoes reach the adult stage. The adults are identified using specific keys and descriptions from the literature (Arnell 1973, Forattini 2002). Mosquitoes belonging to entomologically important species, known to be vectors of etiological agents such as viruses (DENV, ZIKV, YFV), were separated and stored at -80°C for posterior RNA extraction. Therefore, the purpose was to collect eggs from medically important mosquitoes, which are disease vectors such as Dengue, Zika, and Yellow Fever, and not mosquito species with drought-resistant eggs.

References:

Arnell, J. H. 1973. Mosquito studies (Diptera, Culicidae) XXXII. A revision of the genus Haemagogus. Contrib Am Entomol Soc. 10: 1–174.

Forattini OP. Culicidologia Médica.Volume II. São Paulo: Editora da Universidade de São Paulo; 2002.

Moriya, K. 1974. Seasonal trends of field population of mosquitoes with ovitrap in Kanagawa Prefecture: 1) comparison of the populations of four residential areas in Kamakura city in 1971. Jap. J. Sanit. Zool. 25: 237–244.

Navarro, J. C., and C. Machado-Allison. 1995. Aspectos ecológicos de Sabethes chloropterus (Humboldt) (Diptera: Culicidae) en un bosque humedo del Edo. Miranda, Venezuela. Bol. Ent. Venez. (N.S.) 10: 91–104.

Ritchie, S. A., S. Long, G. Smith, A. Pyke, and T. B. Knox. 2004. Entomological investigations in a focus of dengue transmission in Cairns, Queensland, Australia, by using the sticky ovitraps. J. Med. Entomol. 41: 1–4.

Tsuda, Y., M. Takagi, and Y. Wada. 1994. Ecological study on mosquito communities in tree holes in Nagasaki, Japan, with special reference to Aedes albopictus (Diptera: Culicidae). Jap. J. Sanit. Zool. 45: 103–111.

Yanoviak, S. P. 1999. Effects of leaf litter species on macroinvertebrate community properties and mosquito yield in Neotropical tree hole microcosms. Ecologia. 120: 147–155.

L218-219. The authors should describe more how they collect eggs without forcing the readers to read a book and two papers. The size of the container not only the volume. How much of water, what kind of water was put or only rain water? We can not know any detail with the present description.

Answer: We appreciate it and found the suggestion genuinely relevant. For egg collection, we used ovitraps consisting of a 1L-capacity black container without a lid that resembled a plant vase, which contained four wooden oviposition paddles (2.5 cm × 14 cm), vertically held inside the trap by a clip. We added natural water and remains of leaves and animals in each ovitrap in an effort to recreate a microecosystem resembling natural ones.

We have corrected the sentence in the manuscript as follows:

“Monitoring was performed using oviposition traps consisting of a 1L-capacity black container without a lid that resembled a plant vase, which contained four wooden oviposition paddles (Eucatex plates) measuring 2.5 cm × 14 cm, vertically held inside the trap by a clip. Natural water and remains of leaves and animals found on the forest soil were added into the trap to generate an ecosystem similar to the natural one.”

References

Please describe species names in Italics.

Answer: Thank you for pointing that out. We have corrected species names in the references as suggested.

There are many places where words that should be lowercase in the paper titles are in uppercase. Please check every references carefully and make the style uniform.

Answer: Thank you, we corrected the references’ style.

Reviewer 2 Report

The topics of this paper - circulation of ZIKA (and YFV) virus among sylvatic vectors in Brazil - is very interesting and is, for ZIKV, of prime importance for public Health in South America.

However the molecular detection of viral nucleic acid raises concerns :

1) a positive result must be confirmed, the best would be virus isolation. At least a second PCR targeting another viral genomic zone on postive specimens is essential. In my view, the sequencing of the positive amplicon is not sufficient, especially in absence of sequence of the positive controle (and not only the published sequence of the virals train) which could be a cause of cross-contamination.

2) The presentation of the results is not complete : 924 mosquitoes were dispatched in 70 pools when 3662 emerge : what was the rationale for the choice. What species were not tested ?

3) The discussion and the conclusion are not neutral. An example : line 138-139:  "Hg janthinomys is an important canopy vector of YFV, and therefore  should be considered a highly probable ZIKV vector". I don't think there is any dat int he paper to claim this. More, an important paper in this topics is not cited  : DOI: 10.1038/s41598-019-56669-4.

There is not any reference about the vertical transmission where it is a critical point of the paper.

The discussion should be re-drafted with a step-back in the reading of the results, not losing the big picture and be more convincing and less over-claiming.

In the same way the paper should be re-titled like "viral acid nucleic detection in..." rather than "natural infection in .. "

Author Response

We are thankful for the comments and suggestions from anonymous reviewers that helped improve our manuscript. The new version of this manuscript has been read and edited by an English-speaking collaborator.

Reviewer: 2

We would like to thank Reviewer #2 for the essential and valuable comments that made the revised version of the manuscript clearer.

Review Report Form

Open Review

(x) I would not like to sign my review report

( ) I would like to sign my review report

English language and style

( ) Extensive editing of English language and style required

( ) Moderate English changes required

(x) English language and style are fine/minor spell check required

( ) I don't feel qualified to judge about the English language and style

Yes      Can be improved        Must be improved      Not applicable

Does the introduction provide sufficient background and include all relevant references?

(x)       ( )        ( )        ( )

Is the research design appropriate?

( )        (x)       ( )        ( )

Are the methods adequately described?

( )        (x)       ( )        ( )

Are the results clearly presented?

( )        ( )        (x)       ( )

Are the conclusions supported by the results?

( )        ( )        (x)       ( )

Comments and Suggestions for Authors

The topics of this paper - circulation of ZIKA (and YFV) virus among sylvatic vectors in Brazil - is very interesting and is, for ZIKV, of prime importance for public Health in South America.

However the molecular detection of viral nucleic acid raises concerns:

1) a positive result must be confirmed, the best would be virus isolation. At least a second PCR targeting another viral genomic zone on postive specimens is essential. In my view, the sequencing of the positive amplicon is not sufficient, especially in absence of sequence of the positive controle (and not only the published sequence of the virals train) which could be a cause of cross-contamination.

Answer: Thank you for pointing this out. We did not perform virus isolation for the following reasons:

The arbovirus isolation must be performed in a biosafety level 3 laboratory, as shown in the references below. Unfortunately, we do not have this structure in our facility. [Schweitzer BK, Chapman NM, Iwen PC, Abmm D. Overview of the Flaviviridae With an Emphasis on the Japanese Encephalitis Group Viruses. 2009;40: 493–499. doi:10.1309/LM5YWS85NJPCWESW– Bae HG, Nitsche A, Teichmann A, Biel SS, Niedrig M. Detection of yellow fever virus:

A comparison of quantitative real-time PCR and plaque assay. J Virol Methods. 2003;110: 185–191. doi:10.1016/S0166-0934(03)00129-0];

Viral isolation demands from 2 days to one week for viral confirmation. It is time-consuming and not much sensitive, especially if you use immune assays for confirmation [Bae HG, Nitsche A, Teichmann A, Biel SS, Niedrig M. Detection of yellow fever virus:

A comparison of quantitative real-time PCR and plaque assay. J Virol Methods.

2003;110: 185–191. doi:10.1016/S0166-0934(03)00129-0, de Souza Costa MC, Siqueira Maia LM, Costa de Souza V, Gonzaga AM, Correa de Azevedo V, Ramos Martins L, et al. Arbovirus investigation in patients from Mato Grosso during Zika and Chikungunya virus introdution in Brazil, 2015–2016. Acta Trop. Elsevier; 2019;190: 395–402. doi:10.1016/j.actatropica.2018.12.019];

Furthermore, molecular techniques are more efficient and sensitive than virus isolation since RNA degrades rapidly, the amplified material can detect the presence of the virus without virus isolation. Several published articles have used molecular methods with different protocols for the molecular investigation of flavivirus based on the NS5 region:

[Nunes MRT, de Souza WM, Savji N, Figueiredo ML, Cardoso JF, da Silva SP, et al. Oropouche orthobunyavirus: Genetic characterization of full-length genomes and development of molecular methods to discriminate natural reassortments. Infect Genet Evol. Elsevier; 2019;68: 16–22. doi:10.1016/j.meegid.2018.11.020, Bae HG, Nitsche A, Teichmann A, Biel SS, Niedrig M. Detection of yellow fever virus: A comparison of quantitative real-time PCR and plaque assay. J Virol Methods. 2003;110: 185–191. doi:10.1016/S0166-0934(03)00129-0, Mardekian SK, Roberts AL. Diagnostic Options and Challenges for dengue and Chikungunya Viruses. Hindawi Publishing Corporation; 2015;2015.doi:10.1155/2015/834371].

It is also worth mentioning that the NS5 region is considered a necessary enzyme for viral replication. For this reason, it has been used on phylogenetic studies and confirmation of viral particles:

 [Nunes MRT, de Souza WM, Savji N, Figueiredo ML, Cardoso JF, da Silva SP, et al. Oropouche orthobunyavirus: Genetic characterization of full-length genomes and development of molecular methods to discriminate natural reassortments. Infect Genet Evol. Elsevier; 2019;68: 16–22. doi:10.1016/j.meegid.2018.11.020; de Souza Costa MC, Siqueira Maia LM, Costa de Souza V, Gonzaga AM, Correa de Azevedo V, Ramos Martins L, et al. Arbovirus investigation in patients from Mato Grosso during Zika and Chikungunya virus introdution in Brazil, 2015–2016. Acta Trop. Elsevier; 2019;190: 395–402. doi:10.1016/j.actatropica.2018.12.019 6. Morales MA, Fabbri CM, Zunino GE, Kowalewski MM, Luppo VC, Enría; Patel P, Landt O, Kaiser M, Faye O, Koppe T, Lass U, et al. Development of one-step quantitative reverse transcription PCR for the rapid detection of flaviviruses. Virol J. Virology Journal; 2013;10: 58. doi:10.1186/1743-422X-10-58].

We tested all of the collected pools, and most of them had negative results for flavivirus molecular investigation. Besides, we did not include the qRT-PCR results, as all the samples that had negative amplification using RT-PCR also did not amplify when using this technique.  When we performed qRT-PCR and high-resolution melting (HRM), we used cDNA of Yellow Fever 17D, West Nile, and Zika virus as positive controls, with the following melting temperatures: 83.4oC, 79.3oC, and 75.1oC, respectively. Yellow Fever 17D belongs to a vaccinal sample, and West Nile and Zika came from different patients, one with West Nile and the other with Zika disease. All were kindly donated by Dr. Renata Campos Azevedo from Instituto de Microbiologia Professor Paulo de Góes, UFRJ. The Ct’s for those positive samples were 20.3 (YFV 17D), 23.7 (WNV), and 22.0 (ZKV). Finally, the negative template control (NTC) had no signal detected during the 35 cycles used in the qRT-PCR.

To confirm our findings, only RT-PCR positive samples had been purified and sequenced by the Sanger methodology. We also sequenced our positive controls before sequencing our samples. The sequences of the positive controls and positive samples were aligned using Geneious V10.2.4 software, resulting in the mean of 23.6% of identical sites and 82.3% of pairwise homology.

After analyzing the chromatograms, we used the standard nucleotide Blast tool (https://blast.ncbi.nlm.nih.gov/Blast.cgi?PROGRAM=blastn&PAGE_TYPE=BlastSearch&LINK_LOC=blasthome) to confirm our findings.

The pool sizes varied because our technique's minimum RNA detection capacity is for two insects and the maximum capacity for each column supplied in the MN Nucleo Spin® RNA kit is between 30 and 35 mosquitoes.

Our main objective in this study is to carry out epidemiological surveillance monitoring of the presence of flaviviruses in forest and peri-urban areas.

We were also planning to amplify another genomic region. However, the first specific primer set we bought wasn’t amplifying the target genome, and the new primers haven’t arrived yet at our facilities.

The following references have the specific primers for Zika, West Nile and Yellow Fever respectively. The first set of primers we asked were only for Zika and Yellow Fever.

Lanciotti, R.S.; Kosoy, O.L.; Laven, J.J.; Velez, J.O.; Lambert, A.J.; Johnson, A.J.; Stanfield, S.M.; and Duffy, M.R. Genetic and Serologic Properties of Zika Virus Associated with an Epidemic, Yap State, Micronesia, 2007 Emerging Infectious Diseases www.cdc.gov/eid. Vol. 14, No. 8, August 2008 DOI: 10.3201/eid1408.080287

Lanciotti, R.S.; Kerst, A.J.; Nasci, R.S.; Godsey, M.S.; Mitchell, C.J.; Savage, H.M.; Komar, N.; Panella, N.A.; Allen, B.C.; Volpe, K.E.; Davis, B.S.; Roehrig, J.T. Rapid Detection of West Nile Virus from Human Clinical Specimens, Field-Collected Mosquitoes, and Avian Samples by a TaqMan Reverse Transcriptase-PCR Assay. JOURNAL OF CLINICAL MICROBIOLOGY, Nov. 2000, p. 4066–4071 Vol. 38, No. 11 0095-1137/00/$04.00+0

Bonaldo, M.C.; Gómez, M.M; Santos, A.AC; Abreu, F.V.S.; Ferreira-de-Brito, A.; Miranda, R.M.; Castro, M.G.C.; Lourenço-de-Oliveira, R. Genome analysis of yellow fever virus of the ongoing outbreak in Brazil reveals polymorphisms Mem Inst Oswaldo Cruz, Rio de Janeiro, Vol. 112(6): 447-451, June 2017 447

2) The presentation of the results is not complete: 924 mosquitoes were dispatched in 70 pools when 3662 emerge: what was the rationale for the choice. What species were not tested?

Answer: The selection of species for viral detection was based on the degree of importance in the arbovirus transmission, and all pools were processed. We kept processing more pools even after sending this work for publication, except for Limatus durhamii and Culex spp. that were directly in the water of the ovitrap. Our work is dedicated to the constant monitoring of arboviral circulation. By doing this, we hope to make epidemiological monitoring and future alerts of arbovirus circulation in the investigated areas.

3) The discussion and the conclusion are not neutral. An example: line 138-139:  "Hg janthinomys is an important canopy vector of YFV, and therefore should be considered a highly probable ZIKV vector". I don't think there is any dat int he paper to claim this. More, an important paper in this topics is not cited: DOI: 10.1038/s41598-019-56669-4.

Answer: Thank you for pointing this out. We now corrected the sentence as follows:

“Mosquitoes in the states of Ceará and Bahia, where ZIKV has been found in mammals, should also be studied. Although a small number of mosquitoes tested negative in the current study, Hg janthinomysis is an important YFV vector in the canopy and could possibly transmit ZIKV. This species was found in an urban forest (Parque Dois Irmãos) in Recife, Pernambuco, Brazil, a city highly endemic to ZIKV, for this reason, other arboviruses (DENV and CHIKV) should similarly be studied for natural infection.”

There is not any reference about the vertical transmission where it is a critical point of the paper.

Answer: Thank you for pointing this out. We now added the following statement in the Discussion to address this topic: “Vertical transmission in vector mosquitoes has already been reported both naturally and experimentally. In 2016, the vertical transmission of the Zika virus in Ae. aegypti larvae was detected for the first time, found in the field under natural conditions.” 

Reference: Costa CF, Silva AV, Nascimento VA, Souza VC, Monteiro DCS, Terrazas WCM, et al. (2018) Evidence of vertical transmission of Zika virus in field-collected eggs of Aedes aegypti in the Brazilian Amazon. PLoS Negl Trop Dis 12(7): e0006594. https://doi.org/10.1371/journal. pntd.0006594

The discussion should be re-drafted with a step-back in the reading of the results, not losing the big picture and be more convincing and less over-claiming.

Answer: Thank you for the consideration. We have made some modifications to make it less over-claiming.

In the same way the paper should be re-titled like "viral acid nucleic detection in..." rather than "natural infection in .. "

Answer: We consider the detection of viral nucleic acid in mosquitoes as equivalent to natural infection. Unless we suppose the material collected is a degenerated residue from viruses formerly present there, which would be an unusual conclusion. Even if this conclusion is accepted, the mosquitoes were previously naturally infected. Thus, we respectfully decline your suggestion.

Round 2

Reviewer 2 Report

Thanks for the extensive rebuttal that you provided. I considered your answers very appropriate.

However I'd underline the problem of confirmation of positive samples still pending, especially as the core of your paper (spillback of ZIKV towards sylvatic foci) is very important : I'd suggest that you include the following sentence from your answer into the results : "We also sequenced our positive controls before sequencing our samples. The sequences of the positive controls and positive samples were aligned using Geneious V10.2.4 software, resulting in the mean of 23.6% of identical sites and 82.3% of pairwise homology.".

You reckoned the importance of that this confirmation step when you wrote : "We were also planning to amplify another genomic region. However, the first specific primer set we bought wasn’t amplifying the target genome, and the new primers haven’t arrived yet at our facilities.". I really think it'd be a gain  to add these results by waiting for a few weeks. It would reinforce your paper and increase your future citations. However, at this time, I consider it is your choice (and the editor's one) to decide. 

The discussion has been improved accordingly.

Author Response

Reviewer: 2

We would like to thank Reviewer #2 for the essential and valuable comments that made the revised version of the manuscript clearer.

Review Report Form

Open Review

(x) I would not like to sign my review report

( ) I would like to sign my review report

English language and style

( ) Extensive editing of English language and style required

( ) Moderate English changes required

( ) English language and style are fine/minor spell check required

(x) I don't feel qualified to judge about the English language and style

Yes      Can be improved        Must be improved      Not applicable

Does the introduction provide sufficient background and include all relevant references?     (x)       ( )        ( )        ( )

Is the research design appropriate?   ( )        (x)       ( )        ( )

Are the methods adequately described?        ( )        (x)       ( )        ( )

Are the results clearly presented?     (x)       ( )        ( )        ( )

Are the conclusions supported by the results?          (x)       ( )        ( )        ( )

Comments and Suggestions for Authors

Thanks for the extensive rebuttal that you provided. I considered your answers very appropriate.

However I'd underline the problem of confirmation of positive samples still pending, especially as the core of your paper (spillback of ZIKV towards sylvatic foci) is very important: I'd suggest that you include the following sentence from your answer into the results: "We also sequenced our positive controls before sequencing our samples. The sequences of the positive controls and positive samples were aligned using Geneious V10.2.4 software, resulting in the mean of 23.6% of identical sites and 82.3% of pairwise homology.".

Answer: Thanks for your suggestion. We incorporated the information "We also sequenced our positive controls before sequencing our samples. The sequences of the positive controls and positive samples were aligned using Geneious V10.2.4 software, resulting in the mean of 23.6% of identical sites and 82.3% of pairwise homology".

You reckoned the importance of that this confirmation step when you wrote : "We were also planning to amplify another genomic region. However, the first specific primer set we bought wasn’t amplifying the target genome, and the new primers haven’t arrived yet at our facilities.". I really think it'd be a gain to add these results by waiting for a few weeks. It would reinforce your paper and increase your future citations. However, at this time, I consider it is your choice (and the editor's one) to decide.

The discussion has been improved accordingly.

Answer: Thank you for this observation. We are grateful for your considerations, and we are sending three files with the alignment obtained in Geneious software. One of the files has the sequences aligned with the Zika virus sequence provided by our collaborator, one with the NS5 region of the Yellow Fever virus aligned with our sequences, and one contains the alignment with all the sequences we obtained with those downloaded from NCBI.

Meanwhile, we are still waiting for the ordered primers to develop the PCR and sequence the product.
